# Design, Development, and Performance Evaluation of a Fertigation Device for Distributing Solid Fertilizer

**Zhiyang Zhang, Chao Chen, Hong Li * and Huameng Xia**

Research Center of Fluid Machinery and Technology, Jiangsu University, Zhenjiang 212013, China; zhangzhiyang94@163.com (Z.Z.); chch3605@ujs.edu.cn (C.C.); xiahuameng@126.com (H.X.)
* Correspondence: hli@ujs.edu.cn

**Abstract:** Solid fertilizers, which have a low operating cost, are widely applied in Chinese crop fields. In order to distribute solid fertilizer through fertigation, an innovative device with a simple structure was designed, which can feed, dissolve, and distribute fertilizer simultaneously. The parameters of the outlet pipe and fertilizer-feeding component were designed, and a preliminary equation for calculating the fertilizer-feeding flowrate was established. Then experiments were conducted to optimize the established equation. Obtained results showed that the deviation between the measured and the calculated feeding flowrate through the optimized equation was about 5%. This ensured that the fertilizer-feeding flowrate can be adjusted accurately. Experiments were also conducted to explore the effect of the working parameters on the fertilization uniformity of the designed device. It was found that as the fertilization time and inlet water flowrate increases, the fertilization uniformity increases but fertilizer concentration decreases. Based on the obtained results, it is concluded that the designed fertigation device outperforms the conventional pressure differential tank that is normally applied to distribute solid fertilizers.

**Keywords:** fertilization equipment; fertilizer-feeding component; fertilization uniformity; pressure differential tank

---

## 1. Introduction

In the past 10 years, fertigation technology has been rapidly developed in China [1,2]. Fertigation is a technology that allows for transporting water-soluble fertilizers to the field using the irrigation system [3,4]. So far, many investigations have been performed on fertigation equipment. Engel et al. [5] designed a plot seeder and fertilizer applicator, which utilized an injection pump to inject the fluid fertilizer to the field. Samosyuk et al. [6] developed a set of equipment for drip irrigation and fertigation of vegetables, to filter the soluble fertilizer and transport it to the desired point. Maleki et al. [7] applied the "on-the-go" soil sensor and proposed a variable rate granular fertilizer applicator. Back et al. [8] investigated the variable-rate fertilizer applicators with an image-based application rate measurement system, which allows for measuring precisely the rate of granular fertilizers applied. Moreover, Sun [9] designed a multi-channel fertigation machine and proposed a sectional forecast control algorithm based on the nutrient dilution model to improve the efficiency and accuracy of the water-nutrient mixing. Furthermore, diverse end-market products for fertigation, including the NutriFlex, NutriFit, and NutriJet irrigation machines (Priva Company, De Lier, The Netherlands), Elgal-Agro system, automatic fertigation machine (Fertiga, Kfar Blum, Israel), and the MICO-MASTER series products (HARDIE IRRIGATION, Welshpool, UK), have been developed in this regard.

Although many systems have been proposed so far, the majority of existing fertigation devices were designed for liquid fertilizers, which are not popular in China, even though the variety and production of liquid fertilizers increased sharply in recent years [10]. Xu et al. [11] anticipated that

the amount of the fertilizer applied by the Chinese agricultural sector was 60.414 million tons in 2016, while the consumption of liquid fertilizers in the same year was only 805,000 tons. This means that solid fertilizer is still the most popular fertilizer in China. Consequently, it is of significant importance to develop fertigation equipment that uses solid fertilizers. Accordingly, many investigations have been conducted about the fertilization devices with solid fertilizer. Wang et al. [12] developed an automatic control device with the variable rate fertilization, allowing to discharge the solid fertilizer with the 6-line soybean precision planter. Chen et al. [13] designed a self-propelled spreader for rice variable-rate fertilization and studied the correlation between the orifice length and manure flowrate. However, direct scattering of solid fertilizer to the field was not in line with the development of the fertigation. Moreover, fertilization devices, which pre-mix the solid fertilizer and water then distribute the produced fertilizer solution by an injection pump, were studied. Baker et al. [14] designed a rolling spoked-wheel and point-injector fertilizer applicator, which can inject fertilizer solutions almost 10 cm below the soil surface. Chen et al. [15] studied fertilizer automated proportioning equipment, which pre-proportion solid fertilizers and then inject fertilizer solutions through a pressure pump. However, in these devices, users should weigh the fertilizer to adjust different concentrations of the fertilizer solution. This is not only a time-consuming and laborious procedure, but also one or more large-scale fertilizer storage tanks should be matched to meet the fertilizer types and fertilizer application amount. Therefore, it is necessary to develop a fertigation device that does not need to pre-mix solid fertilizer.

It is of significant importance to investigate the correlation between the parameters of the fertigation applicators. Nadiya et al. [16] studied the hydraulic performance of a Venturi injector, Dosmatic fertigation, and fertilizer tank, and established a correlation between the differential pressure and the suction rate or motivation flowrate. Moreover, they obtained a sphere of application based on the discharge rate. Manzano [17] conducted experiments to study the effect of structure and installation alternatives on the head loss, and determined the correlation between the flows and injection efficiency of four types of Venturi injector prototypes. Tang et al. [18] studied the influence of manifold layout and fertilizer concentration on the fertilization time and uniformity of the proportional fertilizer pump. Moreover, Manzano [19] conducted experiments to evaluate the structure and characteristics of the Venturi injector and then proposed a method for selecting the Venturi injector in pressurized irrigation. In the aspect of fertigation applicators performance evaluation, uniformity is commonly utilized as an evaluation index. Li et al. [20], Tayel et al. [21], and Fan et al. [22] combined a pressure differential tank, Venturi applicator, and proportional pump with a drip irrigation system, respectively, and studied the fertilization uniformity of different applicators. They found that the proportional pump has the highest fertilization uniformity. These researchers would be especially instructive to investigate the performance of new fertigation applicators.

In the present article, it was intended to design a uniform fertigation device for distributing solid fertilizers, which can feed, dissolve, and distribute the solid fertilizer simultaneously. Then, experiments were conducted to obtain the influence of structural and working parameters of the designed device on its performance. Finally, the performance of the designed device was compared with that of the conventional pressure tank.

## 2. Structure and Working Principle of the Fertigation Device

Figure 1 shows that the structure of the designed fertigation device, which consists of the fertilizer-feeding, water supply, and mixing components. Moreover, the water supply component consists of a flowmeter, direct current (DC)-pump, and inlet pipe, while the mixing component includes a mixing bucket, filter, outlet pipe, electrical conductivity (EC) sensor, and the sewage outlet pipe. The fertilizer-feeding component feeds the solid fertilizer at the desired flowrate. The filter, which is fixed inside the mixing bucket, screens water-insoluble impurities in the solid fertilizer. The inlet-pipe, which passes through the mixing bucket, is extended to the bottom of the filter so that the inlet water can impact the solid fertilizer deposited at the bottom and accelerates the fertilizer dissolution. The EC

sensor is installed in the outlet pipe, which is fixed on top of the mixing bucket and it is used for indirect detection of the fertilizer concentration through the outlet [23,24]. The working process can be divided into two main stages as the following: (1) Fertilizer application stage, where the solid fertilizer in the fertilizer-feeding device falls into the filter, while water is injected to the mixing bucket by a pump. At this stage, the fertilizer and water are mixed in the mixing bucket. When the liquid level in the bucket exceeds a certain value, the fertilizer solution flows out from the outlet to the field. (2) Rinse stage, in which the fertilizer is no longer fed but the inlet water is still supplied. Consequently, the remaining fertilizer in the bucket can be further replaced and the mixing bucket can be cleaned. When these two stages are completed, the remaining solution in the mixing bucket discharges from the sewage outlet pipe.

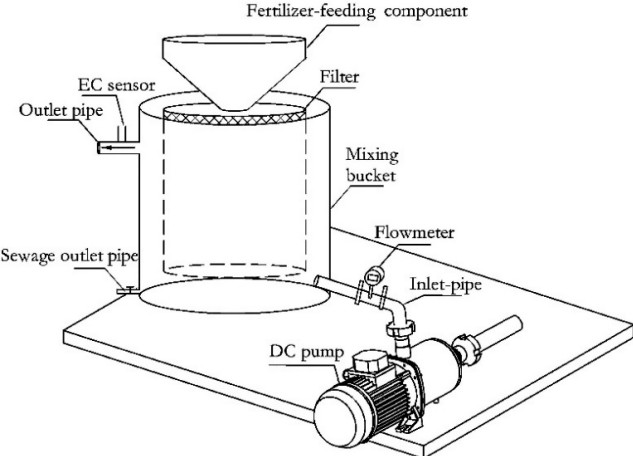

**Figure 1.** Schematic diagram of the fertigation device developed in the current study.

## 3. Designing the Key Components of the Fertigation Device

### 3.1. The Components of the Outlet Pipe

The outlet pipe consists of a cylindrical short nozzle on top of the mixing bucket. It was installed perpendicular to the bucket wall. Moreover, the orifice of the outlet pipe is vertical and sharp-edged. When the fertigation device operates, the fertilizer solution discharges freely through the outlet pipe. The outflow status is shown in Figure 2. It indicates that the fertilizer solution enters the nozzle while it is contracted. Then, it is gradually expanded and finally, it fills the outlet section and flows out. It was observed that the contraction of water cannot extend to the whole outlet section if the pipe is too short. On the other hand, the outlet frictional loss increases as the pipe length increases. Therefore, the outlet pipe length should be 3–4 times the pipe diameter, which is similar to the water outflowing jets [25].

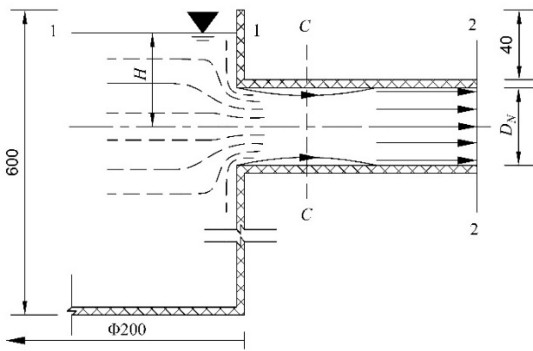

**Figure 2.** Flow analysis at the outlet of the designed device. 1-1 is the water surface section, *H* is the height of water surface from the center of the outlet pipe, *C-C* is the shrink section, 2-2 is outlet section, $D_N$ is the diameter of the outlet pipe.

The pressure of the water surface in the mixing bucket is equal to the atmospheric pressure and the fertilizer discharge process is similar to the cylindrical nozzle outflow. The Bernoulli's equations for 1-1 and 2-2 sections can be expressed as the following:

$$\begin{cases} z_1 + \frac{p_1}{\rho g} + \frac{v_1{}^2}{2g} = z_2 + \frac{p_2}{\rho g} + \frac{v_2{}^2}{2g} + h_w \\ h_w = \zeta \frac{v_2{}^2}{2g} \end{cases},$$

(1)

where $z_1$ (m), $p_1$ (Pa), and $v_1$ (m/s) denote respectively the potential energy, pressure, and velocity in Section 1. Similarly, $z_2$ (m), $p_2$ (Pa), and $v_2$ (m/s) are the potential energy, pressure, and velocity in Section 2. Moreover, $\rho$ (kg/m$^3$) is density of the fertilizer solution, g (m/s) is gravitational acceleration, $h_w$ (m) is the hydraulic loss in the nozzle, and $\zeta$ is the resistance coefficient of the nozzle.

When the device operates, the solution in the mixing bucket rotates slowly, so it is assumed that $v_1 = 0$. Meanwhile, $p_1$ and $p_2$ are equal to atmospheric pressure, and $z_1 - z_2 = H$. Substituting these parameters into Equation (1) results in the following expression:

$$H = (1 + \zeta) \frac{v_2{}^2}{2g}.$$

(2)

Accordingly, the outlet velocity can be calculated as:

$$v_2 = \frac{1}{\sqrt{1 + \zeta}} \sqrt{2gH}.$$

(3)

The resistance coefficient of the nozzle ($\zeta$) is 0.5, so the outlet flowrate can be calculated in the form below:

$$Q_{out} = v_2 A = 0.205 \, \pi D_N{}^2 \sqrt{2gH},$$

(4)

where $Q_{out}$ (m$^3$/h) is outlet flowrate.

It is worth nothing that the volume change of the fertilizer solution in the mixing bucket caused by the dissolution of the solid fertilizer is negligible. Therefore, the inlet water flowrate ($Q$, m$^3$/h) is equal to the outlet flowrate. Accordingly, the following expression can be obtained:

$$H = 0.123 Q^2 D_N{}^{-4}.$$

(5)

When the designed device operates, the liquid level in the bucket remains constant. For water outflowing jets, the water surface is higher than the orifice, which can be mathematically expressed as $H > 1/2 D_N$. Meanwhile, the water surface should not be too high to prevent the solution overflow the mixing bucket. Accordingly, the liquid level in the mixing bucket is set as $H < D_N$. Substituting the abovementioned liquid level into Equation (5) allowed to determine the parameters of the outlet pipe:

$$0.658 \, Q^{0.4} < D_N < 0.755 \, Q^{0.4}.$$

(6)

### 3.2. The Fertilizer-Feeding Component

The fertilizer-feeding component is connected to the cover and it is installed above the mixing bucket as shown in Figure 3. This component consists of a hopper, connector, blanking plate, charging barrel, and a motor. When the fertilizer-feeding component operates, the solid fertilizer stored in the hopper falls into the charging barrel. At the same time, the motor drives the screw rotation, which pushes the fertilizer in the charging barrel down from the side of the blanking plate.

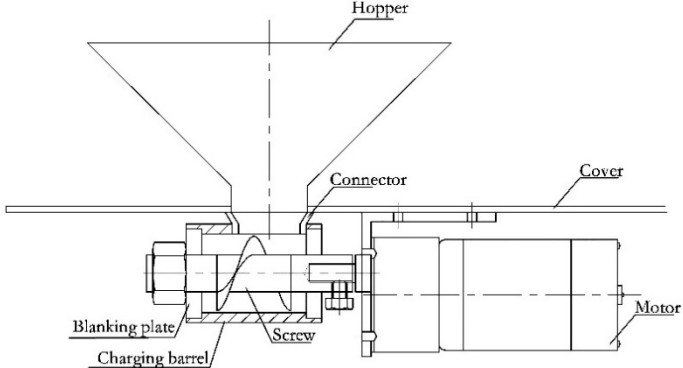

**Figure 3.** Structure diagram of the fertilizer-feeding component in the designed fertigation device.

Ideally, the solid fertilizer fills the gap between the screw and the charging barrel. Therefore, once the screw rotates one turn, the pushed down fertilizer is equal to the filling amount per unit pitch length. For a fertilizer with the bulk density of $\rho$ (kg/m$^3$), motor speed of $n$ (r/min), screw pitch of $L$ (m), screw external diameter of $D$ (m), and screw shaft diameter of $d$ (m), the pushed down fertilizer volume by each rotation of the screw can be calculated through the following expression:

$$V = \frac{1}{4}\pi(D^2 - d^2)L. \tag{7}$$

The corresponding fertilizer-feeding flowrate is:

$$q_m = \rho V n = \frac{1}{4}\pi n \rho (D^2 - d^2)L. \tag{8}$$

## 4. Optimizing the Equation for Calculating the Fertilizer-Feeding Flowrate

### 4.1. Materials and Methods

Based on the structure of the fertilizer-feeding component, Equation (8) was proposed to calculate the fertilizer-feeding flowrate in the optimistic situation of the fertilizer completely filling the gap between the screw and the barrel. However, previous studies showed that this situation does not occur under real conditions [26]. Therefore, it was necessary to modify Equation (8) to establish the exact correlation between the feeding flowrate and the structure or working parameters of the fertilizer-feeding component, so as to accurately adjust the feeding flowrate.

Equation (8) shows that the feeding flowrate depends on the size of the fertilizer-feeding component, fertilizer bulk density, and motor speed. Therefore, two types of fertilizer-feeding components, including size1 ($D$ = 25 mm, $d$ = 12 mm, $L$ = 18 mm) and size2 ($D$ = 50 mm, $d$ = 14 mm, $L$ = 36 mm) were selected. Three types of fertilizers, differing in their bulk densities, were tested, and they were: MOP fertilizer (K$_2$O $\geq$ 60%, bulk density of 1037 kg/m$^3$), compound fertilizer (N + P$_2$O + K$_2$O $\geq$ 40%, N18: P12: K12, bulk density of 924 kg/m$^3$), and urea (total nitrogen $\geq$ 46%, bulk density of 759 kg/m$^3$). It should be indicated that 5 kg fertilizer was added into the hopper, and the motor speed was set to 50, 75, and 100 rpm, respectively. The process started when the fertilizer-feeding component operated and ended when all fertilizer was fed.

### 4.2. Results and Analysis

Table 1 displays the measured feeding flowrate in different conditions. It was observed that the deviation between the measured and calculated flowrate by Equation (8) was high, which was

consistent with the abovementioned analysis. In order to resolve this issue, an additional parameter named the filling coefficient κ was introduced and Equation (8) was optimized into the following form:

$$q_m = \frac{1}{4}\pi\kappa\rho^a(D^2-d^2)^b n^c L^e. \tag{9}$$

**Table 1.** Flowrate of the fertilizer-feeding component of the designed device under different parameters.

| Size of the Screw | Bulk Density of Fertilizer (kg/m$^3$) | Motor Speed (rpm) | $q_m$ | $q_m'$ | $\delta$ | $q_m''$ | $\delta'$ |
|---|---|---|---|---|---|---|---|
| $D = 25$ mm, $d = 12$, $L = 18$ mm | 1037 | 50 | 0.23 | 0.32 | 28.13 | 0.24 | 4.17 |
| | | 75 | 0.32 | 0.47 | 31.92 | 0.35 | 8.57 |
| | | 100 | 0.45 | 0.63 | 28.57 | 0.46 | 2.17 |
| | 924 | 50 | 0.19 | 0.28 | 32.14 | 0.21 | 9.52 |
| | | 75 | 0.28 | 0.42 | 33.33 | 0.31 | 9.68 |
| | | 100 | 0.37 | 0.56 | 33.93 | 0.41 | 9.76 |
| | 759 | 50 | 0.18 | 0.23 | 21.74 | 0.17 | 5.88 |
| | | 75 | 0.26 | 0.35 | 25.71 | 0.26 | 0.00 |
| | | 100 | 0.35 | 0.46 | 23.91 | 0.34 | 2.94 |
| $D = 50$ mm, $d = 14$, $L = 36$ mm | 1037 | 50 | 2.10 | 3.38 | 38.87 | 2.23 | 5.83 |
| | | 75 | 3.26 | 5.07 | 35.70 | 3.28 | 0.61 |
| | | 100 | 4.19 | 6.76 | 38.02 | 4.31 | 2.78 |
| | 924 | 50 | 1.73 | 3.01 | 42.52 | 1.98 | 12.63 |
| | | 75 | 2.54 | 4.51 | 43.68 | 2.91 | 12.71 |
| | | 100 | 3.30 | 6.02 | 45.18 | 3.82 | 13.61 |
| | 759 | 50 | 1.52 | 2.47 | 38.46 | 1.62 | 6.17 |
| | | 75 | 2.36 | 3.71 | 36.39 | 2.37 | 0.42 |
| | | 100 | 3.05 | 4.94 | 38.26 | 3.12 | 2.24 |

where $q_m$ (kg/h) denotes the measured fertilizer-feeding flowrate, $q_m'$ (kg/h) denotes the calculated fertilizer-feeding flowrate through Equation 8, $\delta$ (%) is the relative deviation between $q_m$ and $q_m'$. Similarly, $q_m''$ (kg/h) is the calculated feeding flowrate through Equation (10) and $\delta'$ (%) is the relative deviation between $q_m$ and $q_m''$.

The fitting parameters were determined through the experiment and the equation of the optimized feeding flowrate can be expressed as follows:

$$q_m = 0.304\pi\rho^{1.036}(D^2-d^2)^{0.932}n^{0.948}L^{0.964}; \ (R^2 = 0.990). \tag{10}$$

Table 1 compares the measured and calculated flowrates considering Equations (8) and (10). It was found that as the bulk density and motor speed increased, the measured fertilizer-feeding flowrate increased. The obtained result was consistent with Equations (8) and (10). Moreover, the calculated flowrate through Equation (8) was much higher than the measured flowrate. The relative deviation between them was about 32% in average. Therefore, an optimization was needed, resulting in Equation (10). It was observed that the calculated flowrate through Equation (10) was in good agreement with the measured flowrate and their average relative deviation was 5%. Therefore, Equation (10) has higher accuracy in calculating fertilizer-feeding flowrate than Equation (8). For the fertilizer-feeding component with the same size, the feeding flowrate can be adjusted to a set value by changing the motor speed based on Equation (10). Moreover, Equation (10) provided a basis for selecting the size of the fertilizer-feeding component.

## 5. Fertilization Uniformity of the Fertigation Device

### 5.1. Materials and Methods

#### 5.1.1. Effect of the Structural and Working Parameters on the Fertilization Uniformity

The fertilization uniformity is the most important parameter of a fertigation device for evaluating its performance. Accordingly, experiments were carried out to investigate the effect of the structural and working parameters on the fertilization uniformity of the designed device. The working parameters include the fertilizer amount, fertilization time, and inlet water flowrate and the structural parameters include the parameters of the outlet pipe and structural parameters of the fertilizer-feeding component. The parameters of the outlet pipe were selected to maintain the balance between inlet and outlet flowrates. They did not affect the fertilization uniformity of the designed device. Meanwhile, the structure of the fertilizer-feeding component did not affect the fertilization uniformity directly. This is mainly to the variation of the fertilizer-feeding flowrate, which can be determined by two working parameters, including the fertilizer amount and fertilization time. Therefore, the fertilization uniformity of the designed device was directly correlated to the working parameters only.

The performance experiment was conducted in the Sprinkler Hall, Jiangsu University. The arrangement of the experimental device was shown in Figure 4. The flowrate range of the DC pump (DCZB-48, WANLI) was set to 1.0–1.5 $m^3$/h. The diameter of the inlet pipe was 25 mm, while its installation was 100 mm away from the bottle of the mixing bucket. The flowmeter (CKLDG/LDG) had a range of 0–26.5 $m^3$/h and accuracy of ±0.5%. Furthermore, the fertilizer-feeding component with a screw external diameter of 25 mm, screw shaft diameter of 12 mm, and screw pitch of 18 mm was utilized. The motor (86HBS120, BUKE, Shanghai, China) of the fertilizer-feeding component had a speed range of 0–600 rpm. Moreover, the diameter and length of the outlet pipe were 32 and 96 mm, while its center was 450 mm away from the bottle of the mixing bucket. The operating range and accuracy of the EC sensor (RMD-ISEP10, REMOND, Shanghai, China) were 20,000 and 0.01 μS/cm, respectively. The mesh number of the filter was 60 and its diameter and height were 400 mm. The mixing bucket with diameter of 500 mm and height of 500 mm was produced. The diameter and length of the swage outlet pipe were 15 and 45 mm, while its center was 20 mm away from the bottle of the mixing bucket. The fertilizer applied in the experiment was the abovementioned compound fertilizer. The fertilizer amount was measured using an electronic scale (KF-H2, KAIFENG, Yongkang, China), with the operating range of 0–30 kg and the accuracy of 0.001 kg.

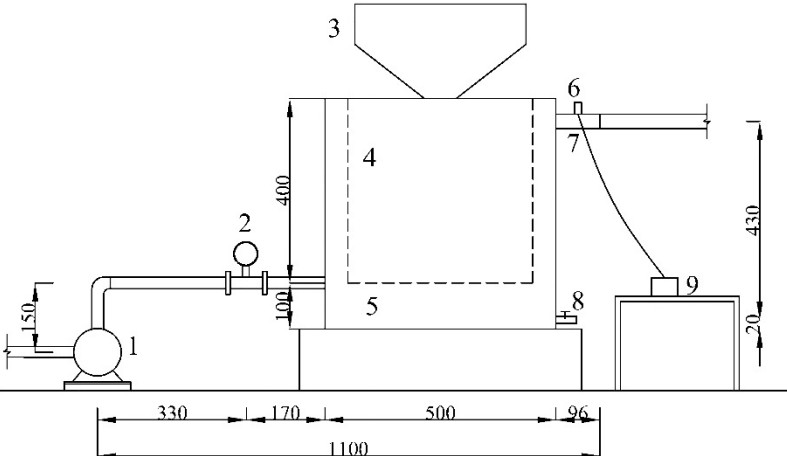

**Figure 4.** Configuration of the experimental device for evaluating the fertilization uniformity of the designed fertigation device. (unit: mm). 1. DC pump 2. Flowmeter 3. Fertilizer-feeding component 4. Filter 5. Mixing bucket 6. EC sensor 7. Outlet pipe 8. Swage outlet pipe 9. EC value display instrument.

The fertilizer-feeding component and DC pump started simultaneously when the designed fertigation device operated. The timer (PC894, TIANFU, Shenzhen, China) started when the fertilizer solution flow out from the outlet pipe. The EC value was recorded every 1 min in the first 20 min of the experiment and then it was sampled every 2 min. In the experiment, six fertilization conditions were covered, which include applying 5 kg fertilizer over 30, 40, and 60 min and applying 10 kg fertilizer over 60, 80, and 120 min. Meanwhile, the inlet water flowrate was adjusted to 1.50, 1.25, and 1.00 m$^3$/h. Therefore, the effect of a single working parameter on the fertilization uniformity of the designed device can be judged.

The following equation was proposed to evaluate the fertilization uniformity of the designed device. It was proposed based on the relative deviation calculation equation, which was used to measure the dispersion degree of data in statistics [27–29]. It should be indicated that the smaller the deviation, the higher the fertilization uniformity of the fertigation device:

$$\delta_F = \frac{\sqrt{\sum_{i=0}^{n} (C_i - C_{\text{ideal}})^2 / n}}{C_{\text{ideal}}} \times 100\%, \tag{11}$$

$$C_{\text{ideal}} = \frac{M}{QT}, \tag{12}$$

where $\delta_F$, $n$, and $C_i$ denote the fertilization concentration deviation, number of measurements, and the $i$-th measured concentration, respectively. Moreover, $C_{\text{idea}}$, $M$, $Q$, and $T$ are the ideal fertilization concentration, fertilizer amount, inlet water flowrate, and fertilization time, respectively.

### 5.1.2. Calibration of the Relationship between Fertilizer Concentration and EC

In the performed experiment, EC values were recorded. However, these values should be converted into the concentration value. Accordingly, it was necessary to calibrate the relationship between the concentration and EC of the compound fertilizer solution involved in this research. In order to improve the accuracy of the measurement, the EC of the fertilizer solution with the same concentration was tested three times. Then, the average value was considered as the final result. Figure 5 shows the relationship between the concentration and EC values.

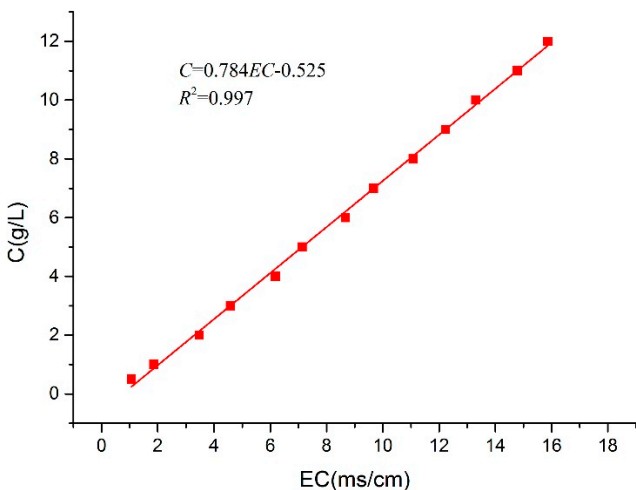

**Figure 5.** Relationship between the concentration and EC values.

The fitted line, using the Origin software, and the obtained regression coefficient were the following:

$$C = 0.784EC - 0.525 \ (R^2 = 0.997). \tag{13}$$

*5.2. Results and Analysis*

　　Figures 6 and 7 show the variation of the fertilization concentration over time under different fertilizer amounts, fertilization time, and inlet water flowrates. The fertilization concentration increased logarithmically in the fertilizer application stage, and decreased exponentially in the rinse stage. Combined with the ideal concentration calculated by Equation (12), it was found that the actual fertilization concentration in the fertilization stage was lower than the ideal concentration. This was because the dissolution of the solid fertilizer had a hysteresis. However, it gradually approached the ideal fertilization concentration over time. Moreover, as the inlet water flowrate increased and the fertilization time decreased, the fertilization concentration and rinse time increased, which was consistent with Equation (12). For example, when the inlet water was 1.50 m$^3$/h, while the fertilizer amount was 5 kg and the fertilization time was 30 min, the average fertilization concentration was 0.72 and 1.65 g/L higher than that 1.25 and 1.00 m$^3$/h. When the fertilization time was 60 min, while the fertilizer amount was 10 kg and inlet water flowrate was 1.50 m$^3$/h, the average fertilization concentration was 0.39 and 1.25 g/L higher than that in 80 and 120 min. Furthermore, with the increase of the fertilizer amount and decrease of the inlet water flowrate, the rinse time decreased. The maximum rinsing time was about 50 min, while the fertilizer amount was 10 kg and the inlet water flowrate was 1.00 m$^3$/h. The minimum rinsing time was about 28 min, while the fertilizer amount was 5 kg and the inlet water was 1.50 m$^3$/h. The inlet water flowrate and fertilizer amount were set by users, so the rinse process cannot be avoided and reduced.

　　Table 2 presents the fertilization concentration deviation of the fertigation device. It was observed that the abovementioned deviation was between 30.37% and 59.59%. Moreover, it was found that as the inlet water flowrate and fertilization time increased, the abovementioned deviation decreased. For example, when the inlet water flowrate increased from 1.0 to 1.5 m$^3$/h, while the fertilization time and fertilizer amount were 30 min and 5 kg, the fertilization concentration deviation decreased by 5.37%. When the fertilization time increased from 60 to 120 min, while the fertilizer amount and inlet water flowrate were 10 kg and 1.25 m$^3$/h, the fertilization concentration deviation decreased by 13.82%. This indicated that the lower the fertilization concentration, the higher the fertilization uniformity of the fertigation device. Furthermore, under the same requirement of the fertilization concentration, the greater the fertilization amount, the smaller the concentration deviation. Therefore, the device obtained a higher fertilization uniformity when applying a large amount of fertilizer over a long time.

**Table 2.** Fertilization concentration deviation (%) of the designed device under different working parameters.

| Flowrate (m$^3$/h) | 5 kg | | | 10 kg | | |
|---|---|---|---|---|---|---|
| | **30 min** | **40 min** | **60 min** | **60 min** | **80 min** | **120 min** |
| 1.50 | 54.22 | 48.18 | 37.57 | 46.52 | 39.13 | 30.07 |
| 1.25 | 55.00 | 51.51 | 44.84 | 47.92 | 42.95 | 35.75 |
| 1.00 | 59.59 | 53.82 | 49.59 | 53.08 | 42.71 | 39.26 |

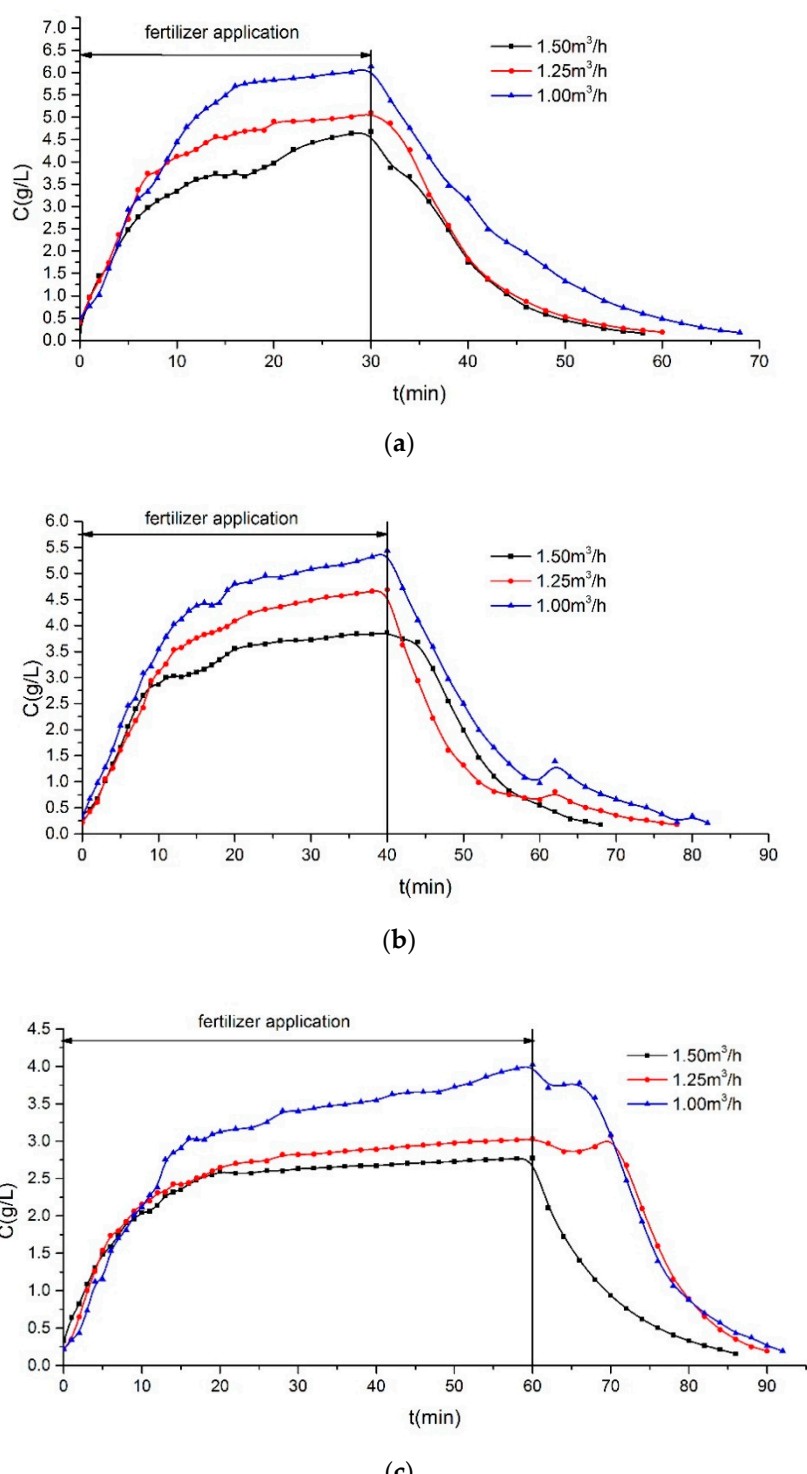

**Figure 6.** Variation of the fertilization concentration with time when the designed device applied 5 kg compound fertilizer. (**a**) T = 30 min, (**b**)T = 40 min, (**c**) T = 60 min.

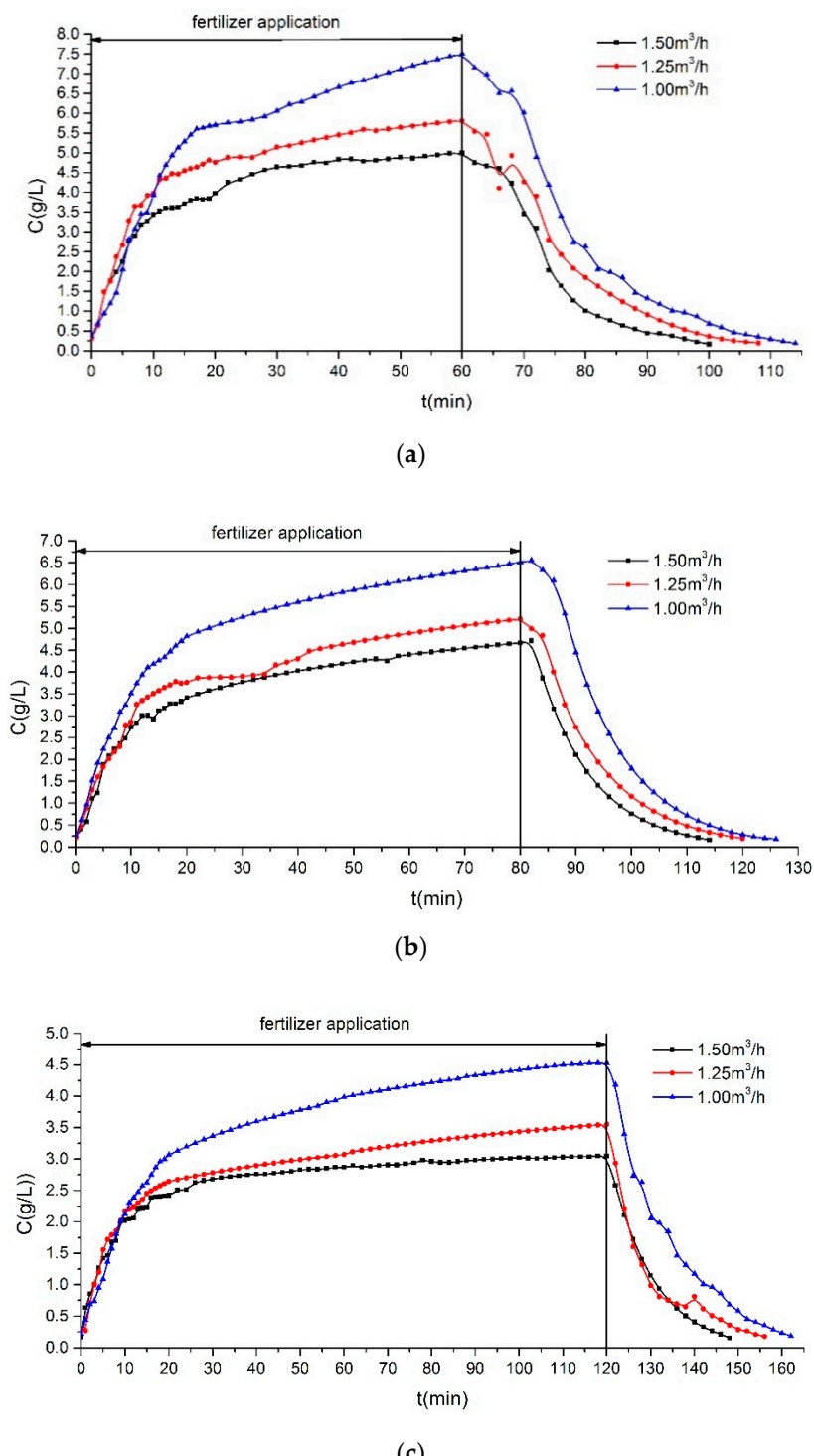

**Figure 7.** Variation of the fertilization concentration with time when the designed device applied 10 kg compound fertilizer. (**a**) T = 60 min, (**b**) T = 80 min, (**c**) T = 120 min.

## 6. Performance Comparison between the Fertigation Device and the Pressure Differential Tank

### 6.1. Materials and Methods

In order to evaluate the hydraulic performance of the fertigation device, a comparison between the designed fertigation device and a pressure differential tank was performed. This comparison was

made because the pressure differential tank applies solid fertilizer similarly to the designed device and is frequently used in field crops.

Figure 8 illustrates the experiment device of the hydraulic performance, the layout of the differential pressure tank, and distances between the elements of the experimental setup. The test water source was from the underground reservoir. Moreover, the outlet of the reservoir was connected with a centrifugal pump (CR10-07, GRUNDFOS, Shanghai, China) to satisfy the flow and pressure requirements of the experiment. The flow and head of the pump were 10 m$^3$/h and 70 m, respectively. It was worth nothing that the valve installed in the head of the test device was utilized to adjust the water flow into the main pipe with the diameter of 50 mm. The flowrate in the main pipe and fertilization pipe were measured by the electromagnetic flowmeter 3 (LWGY-50, Asmik, Hangzhou, China) and flow meter 9 (LWGY-15, Asmik, Hangzhou, China), respectively. Moreover, the accuracy of these two flowmeters was ±0.3%. In order to monitor the pressure difference across the tank, pressure gauges 4 and 6 (YNXC-100, JIANGYUN, Shanghai, China) were installed on the upstream and downstream of the pressure differential tank to monitor the pressure difference across the tank. It should be indicated that the range and accuracy of these two pressure gauges were 0–0.6 MPa and ±0.4%. The pressure difference was regulated by the pressure regulating valve with the diameter of 50 mm. The 50 L pressure differential tank (SFG-50, PEIZE, Shijiazhuang, China) was used in the experiment. The inlet and outlet diameter of the tank was 15 mm. Moreover, the sampling dot 11 was placed in the pressure differential tank outlet, the sample was collected and *EC* was measured by the conductivity meter (CT-3031, JINCHE, Wenzhou, China). The range of the conductivity meter was 0–199.99 mS/cm with the accuracy of ±2% FS.

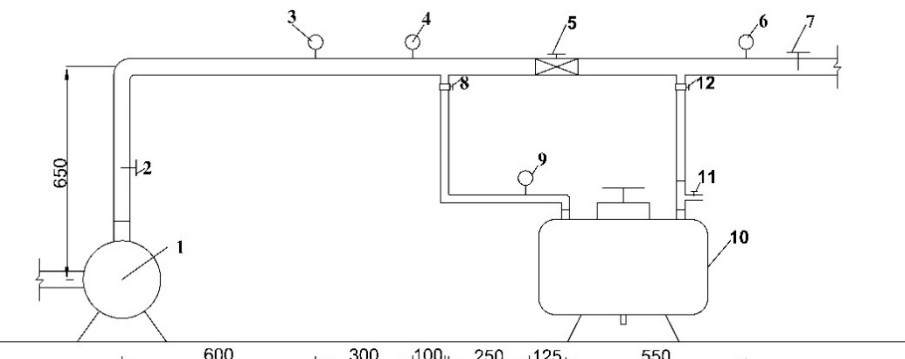

**Figure 8.** Layout of the experiment site for testing the fertilization uniformity of the pressure differential tank. (unit: mm). 1. Pump 2. Valve 3. Flowmeter 4. Pressure gauge 5. Pressure regulating valve 6. Pressure gauge 7. Valve 8. Valve 9. Flowmeter 10. Pressure differential tank 11. Sampling point 12. Valve.

During the experiment, the outlet pressure was maintained at 0.10 MPa, while the inlet pressure was adjusted to 0.15 MPa. Prior to the operation of the pressure differential tank, 5 kg compound fertilizer was added into the pressure tank and then it was dissolved as far as possible. The timer started when the water started flowing through the pressure differential tank. The samples were collected every 1 min in the first 10 min, and then sampling was performed every 5 min.

*6.2. Results and Analysis*

Figure 9 illustrates the performance comparison between the fertigation device and the pressure differential tank. The working condition of the fertigation device was applying 5 kg fertilizer over 30 min, while the inlet flowrate was 1 m$^3$/h. It was found that the concentration of the pressure differential tank rapidly decreased in the initial 10 min and then gradually became stable. Moreover, the concentration of the pressure differential tank decreased from 100 to 0.1 g/L over 70 min, while the concentration change of the fertigation device was only 6.14 g/L. Furthermore, it was observed that

the variation of the concentration curve of the pressure differential tank was sharper than that of the fertigation device. The coefficient of variation of the concentration data of the pressure differential tank was 121.5%, while that of the designed device was 63.7%. Therefore, the fertilization uniformity of the fertigation device was higher than the pressure differential tank.

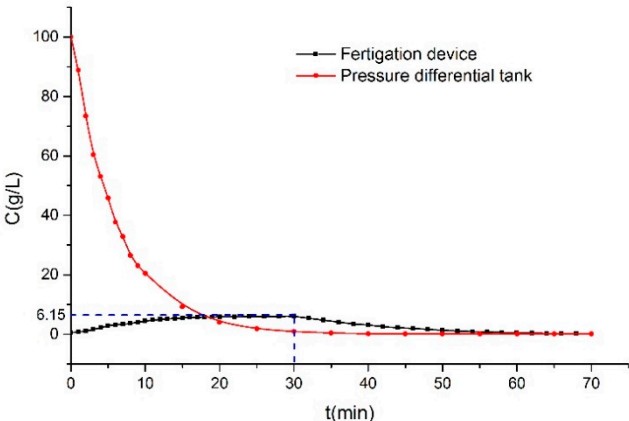

**Figure 9.** Fertilization concentration comparison between the fertigation device and the pressure differential tank.

## 7. Discussion

In the present study, it was found that the fertilization concentration increased logarithmically in the fertilizer application stage. In this stage, the dissolved amount of the fertilizer was more than the output so that the concentration increased. On the other hand, as the fertilization concentration increased, the discharge amount of the fertilizer increased, while the dissolved amount was basically constant. This led to the rapid increment of concentration, which then slowed down. Moreover, it was found that the concentration decreased exponentially in the rinse stage. In this stage, fertilizers were all fed so that there was no fertilizer dissolved. Therefore, the concentration decreased. On the other hand, as the fertilization concentration decreased, the fertilizer discharge amount decreased, which led to the sharp initial decrement of concentration, while slowed down then. Moreover, it was found that the concentration of the pressure differential tank rapidly decreased along time in the initial 10 min and then gradually stabilized. The obtained result was consistent with the study of Li et al. [30]. However, the fertilization concentration deviation of the fertigation device was higher than 30%, which was high for the hysteresis of the fertilizer dissolution. It may further decrease by introducing an agitation unit, which can speed up the dissolution of solid fertilizers. At present, the fertilizer feeding flowrate and the inlet water flowrate were manually calculated and adjusted. In the future, it is hoped that the calculating process can be conducted by a controller. Therefore, the designed fertigation device can operate automatically. It was worth nothing that in this study, a fertigation device was proposed and its performance was evaluated. In further research, it is intended to evaluate the field performance of the proposed device to verify the efficiency and uniformity of the irrigation in combination with the efficiency of nutrient use.

## 8. Conclusions

In the present study, a uniform fertigation device was designed for distributing solid fertilizer. The proposed device can feed, dissolve, and apply the solid fertilizer at the same time. Meanwhile, the fertilization concentration can be adjusted according to the actual requirements. Moreover, the design principles of the outlet pipe and fertilizer-feeding component were presented. The equation for calculating the fertilizer-feeding flowrate was established and then the design is optimized accordingly. Higher accuracy was obtained from the optimized equation, which can guide the precision adjustment of the feeding flowrate. It was found that the fertilization uniformity of the designed device

increases as fertilizer amount, inlet water flowrate, and fertilization time increase. Moreover, it was observed that the fertilization uniformity of the fertigation device was much higher than that of the conventional pressure differential tank. In the proposed device, the fertilization concentration and time of the designed device can be adjusted as required. Therefore, the designed fertigation device can be used in precision agriculture.

**Author Contributions:** Z.Z. and H.X. conceived and structured the testing system. Z.Z. and C.C. performed literature search, helped in experiments and analyzed the test data. Z.Z. and H.L. wrote the paper and approved the submitted version of the manuscript. All authors have read and agreed to the published version of the manuscript.

**Funding:** This work was supported by National key Research and Development Project of China (2017YFD0201502). National Natural Science Foundation of China (51939005). Natural Science Research Project of Jiangsu Higher Education Institutions (19KJB470014).

**Acknowledgments:** A huge thanks is due to the editor and reviewers for their valuable comments to improve the quality of this paper.

**Conflicts of Interest:** The authors declare no conflict of interest.

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
