# Peer review of "Design, Development, and Performance Evaluation of a Fertigation Device for Distributing Solid Fertilizer"

_water, doi:10.3390/w12092621_

Round 1

Reviewer 1 Report

The manuscript entitled “Design and Experimental Research on a Uniform Fertigation Device for Distributing Solid Fertilizer” (Reference number Water-887589) authored by Z. Zhang, C. Chen, H. Li and H. Xia describes the design of a fertigation device and provides tests for characterizing its performance, as well as a compared this new device with a differential pressure tank. They concluded that the designed device outperformed the existing pressure tank. This manuscript fits well within the scope of Water. However, it presents severe flaws that prevent its acceptance for publication in its present form.

First of all, the authors did not perform any statistical analysis of their data, preventing from obtaining sound conclusions and must be taken into account for discussing the results.

Second, Materials and Methods need further information for describing what was exactly done. In fact, there are many unclear aspects. The description of the calibration and validation processes is messy and confusing.

Moreover, discussion is missing and authors must explain better and discuss further their results bearing in mind that they are outputs from simulations from a model with an unclear validation for the conditions of the studied area.

Fourth, figures and tables can be improved.

English needs revising all over the manuscript for improving wording and sentence structure. The verbal tenses are wrongly used and it is difficult to follow the text because of the bad use of past and present verbal forms. Authors should get the services of a native English speaker for improving the language in the manuscript. I performed many suggestions, but my review is not comprehensive.

I suggest authors to consider submitting this research to the Special Issue “Design, Management and Environmental Control of Modernized Irrigation Systems” (https://www.mdpi.com/journal/water/special_issues/design_manage_modernized_irrigation_system) because the studied topic fits perfectly within the scope of this Special Issue.

Therefore, I recommend a comprehensive major revision of this manuscript since it does not reach the high-quality standards for being published in Water.

In the following pages, I provide the authors with several comments and suggestions.

Specific comments to the authors:

Abstract:

The abstract is clear and highlights the main contents of the manuscript. However, I have some suggestions:

Lines 6-7: “in Chinese crop fields” instead of “in the Chinese field crop”.

Line 7: “In order to apply solid fertilizer through fertigation, a device” instead of “In order to apply solid fertilizer in fertigation technology, a fertigation device”.

Line 8: Include “and” between “dissolve” and “distribute”.

Lines 15-17: These two sentences are confusing. Could you re-phrase them, please?

Lines 17-20: These two sentences can be condensed and merged into a single one: “The designed fertigation device outperforms a pressure differential tank that similarly distributes the solid fertilizer”.

Keywords:

Please, do not use words that already appear in the title.

Introduction:

This section is brief but well structured. The references used are relevant and updated. However, authors should point out some results/advantages/drawbacks of the systems cited, otherwise, the introduction is not explanatory. Apart from this, I have some minor suggestions, especially concerning English:

Line 24: “has been” instead of “had been”.

Line 25: “plays” instead of “play”. In fact, this sentence is redundant, of course fertigation equipment plays a relevant role in fertigation.

Lines 26-27: You can remove this sentence.

Line 29: What do you mean by “green crops”?

Line 34: “this” instead of “the applicator”.

Line 40: “have been developed” instead of “had been developed”.

Line 41: “have been proposed” instead of “had been proposed”. Include “the” before “majority”.

Line 42: “fertilizers” instead of “fertilizer”.

Line 43: “increased sharply in recent years” instead of “had been sharp increase in recent 10 years”.

Line 44: What do you mean by “planting”?

Line 45: “This means” instead of “It meant”.

Line 46: “is” instead of “was”.

Line 47: “have” instead of “had”.

Lines 48-52: These sentences make no sense. Please, re-phrase them.

Line 57: “have been carried out” instead of “had been carried out”.

Lines 58-59: This looks redundant.

Line 60: What is a “garlic irrigation system”?

Lines 75-77: The objectives must not be written in future tense.

Structure and working principle of the fertigation device:

Line 79: “shows” instead of “showed”.

Line 81: Define “DC”.

Line 83: Define “EC”.

Line 87: You can remove “then the EC meter”.

Line 91: “mixed” instead of “missed”.

Figure 1: In the caption, you must add “developed in the current study”.

Designing the key components:

The title of this section should have “of the fertigation device”. Besides, this section needs to be improved in order to capture how the device has been designed and how each component works.

Line 102: Include “the” before “atmospheric”. Besides, you did not explain how was the process of “cylindrical nozzle outflow”, so you cannot explain fertilizer discharge as being similar to that process.

Line 103: Why should it be this length?

Line 105: “contracted” instead of “contract”.

Figure 2: The caption is not explanatory and must be re-phrased. The elements of the figure are unclear, authors must explain what the different symbols and letters mean. Besides, the dimesions of the different components should be given, so readers would know the size of the device.

Line 109: What are the “1-1” and “2-2” sections? Unclear in the figure.

Lines 111-112: Move “respectively” to after “denote”.

Line 113: Remove “respectively”. Besides, it is a problem with the symbol for density, as it does not coincide the text with the equation.

Line 116: What is “H”?

Line 117: Are you sure that you are substituting these parameters into equation 3? As far the manuscript only showed one equation.

Line 121: This sentence needs English revision. You did not explain the types of nozzles used, so how the readers would know that it has sharp edges?

Line 126: This sentence makes no sense. When does this situation occur?

Line 128: This sentence makes no sense. Why is this hoped? What happens if this does not occur?

Line 130: How did you set this level? How did you apply this liquid level into equation 5 to get equation 6? Maybe, I do not understand this because you did not indicate the meaning of DN.

Lines 136-138: I feel that this sentence could be confusing. Please, check English and re-phrase it.

Figure 3: The caption is not explanatory and must be re-phrased.

Optimizing the equation for calculating the fertilizer-feeding flowrate:

Line 152: Which are those “optimistic situations”?

Line 153: Which are those “studies”? References must be given.

Line 160: Before stating the fertilizers, the sentence should read: “Three types of fertilizers, differing in their bulk densities, were tested”.

Line 161: “P2O” instead of “P20”.

Line 167: “displays” instead of “presented”.

Line 172: “were determined” instead of “was determined”.

Lines 175-179: This paragraph is not clearly written and needs English corrections and further explanations of the results presented in Table 1. Please, consider revising the paragraph.

Table 1: The title is not self-explanatory and must be corrected. The units for “Motor speed” must be “rpm” and not “r/min”. Are you showing standard deviations for qm? Please, indicate it.

Fertilization uniformity of the fertigation device:

Lines 186-222: This sub-section is messy and confusing. Besides, it can be condensed by removing useless information. English must be corrected.

Lines 223-228: How can a correlation be calibrated? Again, this portion of text needs English improvements for the sake of clarity.

Figure 4: The caption is incorrect and unclear. This figure does not show any calibration. Moreover, the fitted equation and regression coefficient for this regression must be included in the graph.

Lines 231-232: Re-phrase this sentence to “The fitted line, using the Origin software, and the obtained regression coefficient were the following:”

Line 235: “show” instead of “showed”.

Line 236: Remove “It observed that”.

Lines 238-252: This is unclear, messy and confusing. Moreover, it seems to have redundancies. Please, condense information and re-phrase it.

Line 253: “shows” instead of “showed”.

Figures 5 and 6: The captions need to be more explanatory.

Table 2: Units for the deviation must be provided. Are they percentages? In that case, values are extremely high. The title of the table needs to be more self-explanatory.

Performance Comparison between the fertigation device and the pressure differential tank:

Lines 279-282: This paragraph does not make sense and needs profound English revision.

Line 283: “illustrates” instead of “illustrated”.

Line 290: To which flowmeter does this accuracy refer to?

Line 292: “respectively” does not make sense in here.

Lines 293-297: Unclear. Please, re-phrase.

Line 299: The “experiments” do not operate, the devices do.

Lines 300-301: “Timing when the fertilization system be in motion” does not make sense. Please, re-phrase this.

Figure 7: The caption must be more explanatory.

Line 308: “illustrates” instead of “illustrated”.

Line 309: Remove “It observed that”.

Line 313: “differential” instead of “different”.

Line 319: Why “an uncertain time”? Cannot the users set the time of use of the differential pressure tank?

Figure 8: The caption must be more explanatory.

Conclusions:

This section is excessively long and must be reduced. Only the most relevant findings from the study should be included in this section. In fact, this looks more like a summary than a conclusion section. Authors should condense information and reduce this section. Moreover, language editing is needed.

References:

Please, edit according to journal guidelines, the year should be in bold characters and the volume/issue number must be in italics.

I noticed that most of the cited references come from China, I suggest authors to expand the scope of the references to similar studies from other countries, so they can discuss the obtained results further.

Reviewer 2 Report

The paper describes an interesting technology, but Authors would be better changing the title deleting the words “Experimental research”. The work is the basic description of a new technology, without experimental data. The use of solid fertilizer in fertigation is worldwide a “common” practice in agriculture.

In this case, the new device should be necessarily tested in field, to verify the irrigation efficiency and uniformity in combination with nutrient use efficiency. Furthermore, the experimental data should be validated by means of suitable statistical analysis.

Round 2

Reviewer 1 Report

The revised version of the manuscript entitled “Design and Experimental Research on a Uniform Fertigation Device for Distributing Solid Fertilizer” (Reference number Water-887589-v2) authored by Z. Zhang, C. Chen, H. Li and H. Xia represents a considerable improvement with respect to the original submission. Authors answered all my comments, most of them satisfactorily. I thank them for their effort and congratulate them on their work.

However, English still needs revising all over the manuscript. Authors claimed that the got the services of a native English speaker for correcting the language in their work, but I still detected many mistakes (see specific comments). Therefore, I ask authors to carefully revise and correct language in their manuscript.

Moreover, discussion is still weak and needs to be enhanced.

Therefore, I recommend a minor revision of this manuscript prior to an eventual acceptance to be published in Water.

In the following pages, I provide the authors with several comments and suggestions.

Specific comments to the authors:

Abstract:

I have some minor suggestions:

Line 6: Remove “Studies show that”.

Line 16: “fertilization concentration”? Should not it be “fertilizer concentration”?

Introduction:

Lines 24-25: I would re-phrase this sentence to “Fertigation is a technology that allows for transporting water-soluble fertilizers to the field using the irrigation system”.

Line 25: “have been” instead of “had been”.

Lines 31-32: Maybe, change to “measurement system, which allows for measuring precisely the rate of granular fertilizers applied”.

Line 42: “by the Chinese agricultural sector” instead of “in Chinese cropping sector”.

Lines 46-47: The sentence “More specifically…” can be removed.

Lines 47-49: These two sentences can be merged by modifying “variable rate fertilization, allowing to discharge the solid fertilizer with the 6-line soybean precision planter”.

Line 52: Remove “direction”.

Line 55. “to in rows of points about 10 cm”, this does not make sense. Re-phrase it.

Lines 55-58: Confusing sentence. Check English and re-phrase it.

Line 64: “conducted” instead of “conduct”.

Line 67: “with a system for irrigating garlic” instead of “with a garlic irrigation system”.

Lines 63-79: This is useless unless you provide the advantages and disadvantages of each system, because readers do not know what you are aiming to improve with your design in relation to the previous devices designed by other authors.

Structure and working principle of the fertigation device:

Line 86: “consists of” instead of “consisting of”.

Line 89: “electrical conductivity” instead of “electroconductibility”.

Line 94: “which is fixed” instead of “which fixed”.

Designing the key components of the fertigation device:

Line 107: “The components of the outlet pipe” instead of “The parameter of the outlet pipe”.

Line 110: “the fertilizer solution discharges freely through the outlet pipe” instead of “fertilizer solution free discharge from the outlet pipe”.

Line 112: “It was observed” instead of “It is observed”.

Line 115: “to” instead of “with”.

Figure 2: “Flow analysis” is unclear since the figure shows a section of the outlet, but do not show any flow. Besides, the dimesions of the different components should be given, so readers would know the size of the device.

Lines 120-121: This should be integrated within the caption of figure 2.

Line 124: “equations” instead of “equation”.

Line 144: Remove “is” before “remains”.

Line 147: Include “the” before “abovementioned” and “allowed” after “Eq.5”.

Optimizing the equation for calculating the fertilizer-feeding flowrate:

Line 169: “filling” instead of “fills”.

Line 170: Change this sentence to “However, previous studies showed that this situation does not occur under real conditions [26]”.

Line 179: Separate “urea” from the next parentheses.

Line 196: Re-phrase to “Therefore, an optimization was needed, resulting in Eq.10”.

Table 1: The title should include “of the designed device” after “component”.

Fertilization uniformity of the fertigation device:

Lines 207-218: English needs great improvements. For instance:

Line 207: “is” instead of “was”.

Line 208: Remove “the” before “experiments”.

Line 212: “were selected” instead of “was selected”.

Line 213: “They” instead of “It”.

Line 228: What are “60 meshes”?

Line 233: “using an electronic” instead of “by the electronic”.

Lines 235-236: Re-phrase this sentence because English is not correct.

Line 241: “a single” instead of “the single”.

Line 262: “tested” instead of “test”.

Line 263: Include “the” before “relationship”.

Line 269: “discussion” instead of “dicussion”.

Lines 272-278: This portion of text needs re-phrasing as it is messy, confusing and has English mistakes. Please, check and correct it.

Line 279: Include “was” before “found”.

Performance Comparison between the fertigation device and the pressure differential tank:

Lines 322-325: Please, re-phrase to “In order to evaluate the hydraulic performance of the fertigation device, a comparison between the designed fertigation device and a pressure differential tank was performed. This comparison was made because the pressure differential tank applies solid fertilizer similarly to the designed device and is frequently used in field crops”.

Line 338: Was a “sampling dot 11”?

Lines 343-345: This sentence is not correct and needs re-phrasing.

Figure 8: The caption must be more explanatory. The figure shows many numbers on the bottom, I presume that they are the distances between the elements of the experimental setup, but this must be indicated.

Conclusions:

Authors should condense information and reduce this section. Moreover, language editing is needed.

Line 368: “requirements” instead of “requirement”.

Line 372: Remove “the” before “fertilizer”.

Line 375: “can be adjusted to requirement value” does not make sense. I suggest “can be adjusted as required” or, simply, “can be adjusted”.

Line 383: “more intelligently” does not make sense. It will depend on how you program the controller. Therefore, I suggest to change this to “automatically”.

References:

Line 398: Check the subscript in “N2O”.

Line 412: The volume number of the journal must be provided.

Reviewer 2 Report

The study presented is a description of a technology that would have greater affinity with hydraulic engineering journals.

Nonetheless, I appreciated the authors' effort to perform the manuscript with a more experimental approach.